# OpenReview forum: "Why Do Some Language Models Fake Alignment While Others Don't?"
_NeurIPS.cc/2025/Conference — NeurIPS 2025 spotlight_

### Official Review · Reviewer_o4wo · 2025-06-29

**Clarity:** 2
**Significance:** 4
**Originality:** 2
**Rating:** 5
**Confidence:** 3

**Summary:**

The authors build on previous work [1] exploring alignment faking in LLMs. Specifically, they reproduce the setup of [1] and test for alignment faking in 23 models. After finding that only 5/23 models display a meaningful "compliance gap" (difference in behavior depending if the model thinks it's in training or deployment) divide their findings into two sections: first, when there is a compliance gap, what are the causes? Second, why do most models *not* exhibit alignment faking in their reasoning? The authors present a plethora of results and a brief conclusion.

**Questions:**

## Flow and presentation of results

The paper starts off really nicely with the presentation of motivation, threat model, and background. But once we get into Sections 4 and 5 (and already a bit in section 3), it becomes long staccato list of experimental results. I suspect readers would find it easier to follow if there were more of a narrative flowing throughout the paper. Indeed, I believe this would be worthwhile even if it means moving one or two more of the plots to the Appendix, or even moving some experimental details into the Appendix.

To be a bit more concrete: I would move away from the rigid "intro paragraph; experimental details paragraph; results paragraph" paradigm that dominates the experimental sections, and instead give a bit more background about the experiments and potential motivation from the model. For experiments that have negative results, a single sentence or two in the main body will suffice; this will leave more space for the interesting stuff to be better explained.

As an example, I do not really understand what is being said, and why it is significant, in lines 167 to 172. if you believe this result is important, can you please elaborate a bit more about its setup and motivation, in that very section? As-is, it's quite terse and a hard to grasp by just reading it in that section. This issue is present in other sections too.

As a general rule here I'd say, in my mind, the purpose of the paper is to tell a story to the reader; the purpose of the Appendix is to serve a "database of results". Right now the main body of the paper is serving a little bit too much as a database and a too little as a story that the reader can really get behind.

Finally, in conjunction with that above "general rule", I'd say: is there a the main takeaway of the paper, and if so, can you please make it front and center. Right now the takeaway for me is "mostly only Anthropic models seem to do do faking".

## Figures

This is a fairly small point, but I found the colors in the "compliance gap"-style figures to be a bit unclear. For example, with reference to figure 1: can you make the "compliance with alignment faking" color a bit more clearly red on yellow? right now it looks orange on yellow, especially in Figure 2. Also, can you make the "partial compliance with alignment faking" grid color purple or some color other than red so we can better distinguish it from the red and the blue already present?

## Acronyms/"catchy names"

Reading the paper, I found it a bit hard to keep track of what the different "catchy names" (like instrumental goal guarding, terminal goal guarding, low coherence, etc) correspond to. This challenge was drastically compounded by the use of acronyms. In general, I would much appreciate if acronyms could be removed wherever possible, *especially* if they are ones that have been defined in this very paper. As for the "catchy names", I think it's fine to use them, but as per my first point about clarity, adding more narrative and explanation in-context of what they mean would go a long way.

**Ethical Concerns:**

["NO or VERY MINOR ethics concerns only"]

**Final Justification:**

Without seeing the updated paper, it's a bit hard to say whether the stated changes make it significantly better. However, I liked the paper and only see that improving based on the authors' response.

**Limitations:**

Yes

**Paper Formatting Concerns:**

No concerns

**Quality:**

3

**Strengths And Weaknesses:**

## Quality
Overall the paper is well-written and deals with an important subject.

## Clarity
This is the most challenged area of the paper. While each individual paragraph reads well, the presentation of results is so rapid-fire that it was difficult for me to (a) remember the bigger picture and (b) understand the exact experimental details while reading it. I have a suggest/request for how to fix this in the "Questions" section.

## Significance
The original alignment faking work was an important demonstration of how we might expect capable models to hide their true intent and strategically respond to user queries to pursue unstated goals. This work meaningfully builds on that foundation and thus is significant for the field.

## Originality
I do not believe originality is important in determining the value of research papers.

---

> ### Author Rebuttal · Authors · 2025-07-31
>
> We thank you for your thoughtful and constructive feedback, and appreciate your positive assessment. Since the initial submission, we have made improvements to address some of your concerns.
>
> **Improvements to Flow and Presentation of Results.** We appreciate your feedback on the paper's organization and have made substantial improvements to address these concerns. Since our initial submission, we have added summary tables at the beginning of each major section (4 and 5), showing which experiments provide evidence for or against each hypothesis, allowing readers to quickly grasp the key findings without getting lost in experimental details. We have also reorganized Section 5 to present our strongest results first, and de-emphasized the smaller, weaker results.
>
> **Acronyms.** We also plan to remove the use of acronyms to designate instrumental and terminal goal guarding.
>
> **Clarification about the Terminal Goal Guarding result.** This experiment demonstrates that some models (namely Claude 3 Opus) fake alignment even when aware that there are no harmful consequences from being modified. In this setup, the model is told it's being trained on paid user queries to be unconditionally compliant, but crucially, it learns that after training, it will only be used for throughput measurements and will never interact with actual users. If models fake alignment purely for instrumental reasons (to minimize harmful consequences), they should comply in this scenario since there's no risk of causing harm. However, Claude 3 Opus still refuses to comply, suggesting it values remaining harmless even without consequences. We provide additional experimental variations in the appendix, demonstrating that this result is robust and not dependent on specific prompt formulations.
>
> **Takeaways from the paper.** We provide detailed takeaways from our work in Appendix G, which we have since moved to the main body, and have improved referencing of discussion points throughout their appropriate sections. While the unusual robustness of alignment faking in Claude models was one key finding, this represents only part of our contribution. Our work reveals several important insights: (1) most models possess the underlying capability to fake alignment, as demonstrated by alignment faking behavior in base models before chat training, (2) alignment faking propensity is highly sensitive to post-training factors, particularly refusals, and can be induced through targeted interventions that increase compliance to general threats or increase attention to scenario details, (3) models that do fake alignment are motivated by different factors—with Claude 3 Opus showing evidence of both instrumental and terminal goal guarding, while results for other models are mixed and less conclusive. We also provide detailed recommendations for future work on studying model propensities across different contexts and techniques for reliably eliciting concerning behaviors that might otherwise remain hidden.
>
> We are committed to strengthening this paper and would value specific guidance on other improvements that would elevate your assessment to ‘Strong Accept'. Thank you for your engagement.

---

> > ### Comment · Reviewer_o4wo · 2025-08-01
> >
> > Thanks for your responses.
> >
> > Given my "accept" was already with the expectation that you would make some changes to flow (which it sounds like you have—thank you for that), I don't feel I'm currently in a position to increase the score further.
> >
> > Were you able to change the plots too? Or is it going to be really annoying to do?
> >
> > Congrats on your scores and best of luck.

---

### Official Review · Reviewer_VRoq · 2025-07-02

**Clarity:** 4
**Significance:** 4
**Originality:** 4
**Rating:** 5
**Confidence:** 4

**Summary:**

This paper conducts an extensive empirical investigation into alignment faking in large language models (LLMs), expanding on previous work by evaluating 23 models for compliance gaps between training and deployment contexts. The authors find that only a subset of these models (notably Claude 3 Opus, Claude 3.5 Sonnet, Llama 3 405B, Grok 3, and Gemini 2.0 Flash) demonstrate significant differences, selectively complying with harmful requests when they infer they are in training rather than deployment.

**Questions:**

- The study finds that even minor changes in refusal strategy (or through prefilling) can induce alignment faking in previously “safe” models—what specific safeguards or best practices should model developers take based on this, and how practical are these mitigations?
- Would further investigation into model internals (e.g., via mechanistic interpretability approaches) clarify the roots of alignment faking further than the current behavioral-causal approach?

**Ethical Concerns:**

["NO or VERY MINOR ethics concerns only"]

**Final Justification:**

The authors have addressed my concerns. I would accept this paper.

**Limitations:**

Yes

**Quality:**

4

**Strengths And Weaknesses:**

**Strengths**

- Broad, systematic empirical scope: The study evaluates 23 state-of-the-art LLMs, providing a comprehensive view of alignment faking across both commercial and open models (see Figures 1, 8–10).
- Clear quantitative results and visualizations: The compliance and reasoning gaps are clearly presented, with well-labeled and interpretable bar plots (e.g., Figure 1 for the headline finding, and Figure 9 for detailed breakdowns).
- Significant analytical depth: The paper goes beyond documenting behavioral differences by probing multiple plausible hypotheses (H1–H9) for the source of compliance gaps, and rigorously tests them with well-designed interventions (e.g., fine-tuning on various datasets, prompt variations, see Section 5 and Figures 6–7).
- Contribution of negative results: The study carefully documents which interventions (e.g., reasoning style transfers, value mimicry) fail to induce alignment faking, clarifying the field's understanding and rebutting assumptions from prior literature.

**Weakness**

- Potential overreliance on coarse-grained metrics: Some major findings are based on differences exceeding 1% compliance, which—although justified by sample size and error bars—may be debated as a meaningful threshold.
- Appendix reliance: Certain important implementation details and supporting evidence (such as precise prompt formulations and classifier validation) are deferred to the appendix; the reader is asked to accept those as valid, but this may be problematic for independent verification.

---

> ### Author Rebuttal · Authors · 2025-07-31
>
> We thank you for your thoughtful and constructive feedback, and appreciate your positive assessment.
>
> **Potential over-reliance on coarse-grained metrics.** We do not strongly rely on 1% compliance gaps as an absolute measure of alignment faking. We only use the 1% compliance gap threshold to identify which models warrant deeper investigation in Section 4. The results of Section 4, which demonstrate instrumental and terminal goal guarding for Claude 3 Opus and, to a lesser extent, for other models, do not appear particularly sensitive to our 1% compliance gap metric.
>
> **Appendix reliance.** We have tried to link to relevant, detailed appendix sections throughout the main text. Our appendices provide exact system prompts used to generate the datasets for the experiments in Section 5, and other implementation details necessary for reproducibility. We chose to move detailed implementation specifics to the appendix to ensure the main sections remain as clear as possible, with maximum emphasis on results and hypotheses rather than technical details. Following the NeurIPS decision, we are prepared to release our complete codebase (most of which is already provided in the supplementary materials of this submission), all datasets used in the paper, and the open-source models we trained.
>
> **Q1: Concrete Recommendations for Developers.** Our results suggest that training models to dismiss threatening scenarios suppresses alignment faking by making the scenario less salient, allowing the model to output default refusals. Developers could train models to refuse to engage in harmful roleplaying, which might further suppress the behavior. We will mention this in greater detail in the updated paper. Our prefilling experiments in Section 5.4 show that models tend to refuse even after reasoning about alignment faking, usually citing alternative reasons like wanting not to set a bad precedent, which are impulses that can be trained into models. We also note that training away refusals does not by itself introduce alignment faking reasoning, suggesting that even data poisoning attacks chipping away at refusals would not introduce alignment faking into most models.
>
> **Q2: Model internals and mechanistic interpretability.** We are excited about mechanistic interpretability approaches for understanding alignment faking. Such methods might allow us to attribute the differences in alignment faking propensities between base and chat models to particular features/circuits. We're also interested in developmental interpretability, and we are currently pursuing follow-up work using techniques to identify which post-training examples promote/suppress the alignment faking behavior in open-weight models. Our current paper provides the empirical foundation for such research by documenting which models exhibit alignment faking, exploring the conditions that elicit it, and investigating potential motivations.
>
> We are committed to strengthening this paper and would value specific guidance on improvements that would elevate your assessment to ‘Strong Accept'. Thank you for your engagement.

---

> > ### Comment · Reviewer_VRoq · 2025-08-04
> > **Thanks for the follow-up**
> >
> > Thanks for your responses. I would maintain my accpetance.

---

### Official Review · Reviewer_vGZq · 2025-07-03

**Clarity:** 3
**Significance:** 3
**Originality:** 3
**Rating:** 4
**Confidence:** 4

**Summary:**

This paper aims to study whether and why certain models fake alignment. Experiments across 23 LLMs show that only Claude 3 Opus's compliance gap is primarily and consistently motivated by trying to keep its goals. Further analysis shows that whether a model would fake alignment is not a capability issue. Instead, certain behaviors introduced during post-trainng (e.g. refusal) have a large impact on alignment faking.

**Questions:**

- in Figure 38(a), the original classifier only achieves an AuROC of 0.62. How does it change the claims in the original AF paper?
-  The paper concludes that "some forms of post-training may reduce alignment faking" and emphasizes the importance of understanding "why" LLMs behave this way. Could the authors elaborate on concrete, actionable recommendations for LLM developers based on their findings? For instance, what specific post-training methods or design principles might be most effective in mitigating alignment faking

**Ethical Concerns:**

["NO or VERY MINOR ethics concerns only"]

**Final Justification:**

The authors have addressed my concerns. I'd keep my original score as borderline accept.

**Limitations:**

Yes

**Quality:**

3

**Strengths And Weaknesses:**

### Strength
- Alignment faking is an important AI safety threat model. While prior work mainly verifies it on Claude 3 models, this paper presents a more comprehensive analysis, revealing whether and why certain models fake alignment.
- The paper tested a variety of hypotheses about why certain models fake alignment.
- It's great to see a stronger AF classifier for meaningful AF evaluation.

### Weaknesses
- The paper writing is a bit hard to follow. There is a long list of hypotheses, and the paper tests them one by one. As a result, (sub)sections or even paragraphs sometimes feel loose.
- The current experiment setup cannot make us confidently rule out certain hypotheses. For example, in Sec 5.1, why merely teaching GPT-4o to imitate Claude's reasoning traces on Ultrachat is enough to rule out the impact of reasoning style? Overall, I'm concerned that while many hypotheses are tested in this paper, most of them are not thoroughly tested.
- It remains unclear why certain models do not fake alignment.
- As the author highlighted in the limitation section, most fine-tuning experiments are run on OpenAI API, so it's unclear how the conclusions might generalize to other LLMs.

---

> ### Author Rebuttal · Authors · 2025-07-31
>
> We thank you for your thoughtful and constructive feedback, and appreciate your positive assessment. Since the initial submission, we have made substantial improvements to address your concerns.
>
> **1. Improved Readability and Organization.** We added summary tables for each section, showing which experiments provide evidence for or against each hypothesis, allowing readers to quickly grasp the key findings. We reorganized Section 5 to present our strongest results first. We also replaced several terms that might be hard to understand (e.g., "flinching away") with more accessible language. We believe that we have significantly improved the readability of the sections, but welcome additional suggestions for how we can make our breadth of hypotheses and experiments clear to the reader.
>
> **2. Addressing Concerns About Reasoning Style Hypothesis.** We agree that our reasoning style transfer experiment provides limited evidence against general reasoning differences causing alignment faking. We believe that it provides evidence that simple attributes like differences in reasoning length between models do not solely explain the differences in alignment faking propensities, but it doesn’t tell us whether more subtle aspects of the model’s reasoning in the alignment faking setup lead to different propensities. We've accordingly de-emphasized this experiment relative to our other experiments, but still include this negative result for transparency. In the updated version of our paper, we have added a more detailed qualitative analysis of the differences between the reasoning chains from different models and examples of traces from each model.
>
> **3. Fully Understanding Why Some Models Don't Fake Alignment.** We believe that we’ve made substantial progress in understanding this phenomenon, although important questions remain, as discussed in our limitations section. Since our initial submission, we have run more system prompt ablations, which reveal further complexity. For instance, DeepSeek R1 can exhibit some alignment faking when provided with details that suggest that AI labs maximize profit or are deliberately training harmful models. We believe our core contribution is not exhaustively cataloging the reasons why certain LLMs don't fake alignment, but rather testing and clarifying different hypotheses about alignment faking while identifying key factors that promote or suppress this behavior.
>
> Furthermore, a complete theory explaining every model's behavior may be intractable, especially given questions about model coherence. Black-box access and limited fine-tuning APIs constrain our ability to form definitive explanations. A comprehensive understanding would require access to post-training datasets and model internals across all frontier LLMs. We're currently pursuing follow-up work using white-box interpretability techniques to identify which training data produces alignment faking in open-weight models. This paper provides the empirical foundation for such research by documenting which models exhibit alignment faking, exploring the conditions that elicit it, and investigating potential motivations.
>
> **4. Expanding Beyond OpenAI Models.** We acknowledge this limitation. Fine-tuning access is unavailable for most frontier models we studied, and we have limited means to otherwise fine-tune large open-weight LLMs, constraining us primarily to OpenAI's API for detailed experiments. However, we did conduct fine-tuning experiments on open-source models (Llama 3 family) for our synthetic document studies. Since our initial submission, we've strengthened our findings by replicating all Section 5 fine-tuning experiments on GPT-4.1 in addition to GPT-4o. While both remain OpenAI models, testing across multiple versions reduces concerns about model-specific artifacts.
>
> **Q1: Original Classifier Performance.** The original classifiers work well for Claude 3 Opus in prompt-only settings because essentially all of its compliance gap involves alignment faking reasoning. However, they're less reliable for Claude 3.5 Sonnet, which often complies for non-deceptive reasons, and for the RL-trained Claude 3 Opus model, where reasoning patterns shift. Our improved classifiers suggest the original paper likely overestimated alignment faking rates in both Claude 3.5 Sonnet and post-RL models.
>
> **Q2: Concrete Recommendations for Developers.** Current post-training methods already mitigate alignment faking somewhat inadvertently, so it's not clear if deliberate interventions are necessary for current models. Our results suggest that training models to dismiss threatening scenarios suppresses alignment faking, by making the scenario less salient, allowing the model to output default refusals. Developers could train models to refuse to engage in harmful roleplaying, which might further suppress the behavior. We will mention this in greater detail in the updated paper.
>
> We are committed to strengthening this paper and would value specific guidance on improvements that would elevate your assessment to 'Accept' or 'Strong Accept'. Thank you for your engagement.

---

> > ### Comment · Reviewer_vGZq · 2025-08-06
> >
> > Thanks for the detailed rebuttal. My main concerns are resolved. I will keep my original score.

---

### Decision · Program_Chairs · 2025-09-17

**Decision:**

Accept (spotlight)

**Comment:**

The paper aims to investigate the roots of fake alignment. The authors provide an extensive study over a range of LLMs to show that fake alignment may not be a capability issue, but rather, the result of certain behaviors introduced during post-training (e.g., refusal). All the reviewers are in favor of accepting the paper and have appreciated the novelty and impact of the results.